

# A novel transcript of MEF2D promotes myoblast differentiation and its variations associated with growth traits in chicken

Hongjia Ouyang[1,2], Jiao Yu[3], Xiaolan Chen[3], Zhijun Wang[3] and Qinghua Nie[3]

[1] College of Animal Science & Technology, Zhongkai University of Agriculture and Engineering, Guangzhou, China
[2] Guangdong Province Key Laboratory of Waterfowl Healthy Breeding, Guangzhou, China
[3] Department of Animal Genetics, Breeding and Reproduction, College of Animal Science, South China Agricultural University, Guangzhou, Guangdong, China

## ABSTRACT

**Background:** Development of skeletal muscle is closely related to broiler production traits. The myocyte-specific enhancer binding factor (MEF) 2D gene (*MEF2D*) and its variant transcripts play important parts in myogenesis.

**Methods:** To identify the transcript variants of chicken *MEF2D* gene and their function, this study cloned chicken *MEF2D* gene and identified its transcript variants from different tissue samples. The expression levels of different transcripts of *MEF2D* gene in different tissues and different periods were measured, and their effects on myoblast proliferation and differentiation were investigated. Variations in MEF2D were identified and association analysis with chicken production traits carried out.

**Results:** Four novel transcript variants of *MEF2D* were obtained, all of which contained highly conserved sequences, including MADS-Box and MEF2-Domain functional regions. Transcript *MEF2D-V4* was expressed specifically in muscle, and its expression was increased during embryonic muscle development. The *MEF2D-V4* could promote differentiation of chicken myoblasts and its expression was regulated by *RBFOX2*. The single nucleotide polymorphism g.36186C > T generated a TAG stop codon, caused MEF2D-V4 to terminate translation early, and was associated with several growth traits, especially on early body weight.

**Conclusion:** We cloned the muscle-specific transcript of *MEF2D* and preliminarily revealed its role in embryonic muscle development.

Corresponding author
Qinghua Nie, nqinghua@scau.edu.cn

## INTRODUCTION

The myocyte-specific enhancer binding factor 2 (MEF2) family is widely present in muscle cells. It plays an important role in the development, growth and maintenance of organisms through interacting with various genes in the calcineurin signaling pathway (*Potthoff & Olson, 2007*). MEF2 is a major regulator of myogenic genes expression, which can activate expression of various myogenic related genes, and interact with members of myogenic regulatory factors to regulate myogenesis (*Molkentin et al., 1995*; *Desjardins & Naya, 2016*). In vertebrates, the MEF2 family has four members, including MEF2A, MEF2B,

MEF2C and MEF2D genes. MEF2 belongs to the MADS-Box family of transcription regulators. The N-terminal of the four MEF2 proteins all contain the highly conserved MADS-box domain and MEF2 domain. The structural difference among them is due mainly to the difference in C-terminal transcriptionally active regions (*Molkentin et al., 1996*; *Black & Olson, 1998*). *Breitbart et al. (1993)* first cloned *MEF2D* in humans, and found that it plays a key role in muscle development. As a member of MEF2 family, *MEF2D* has been reported that plays a key role in myogenesis. In *MEF2D* knockout mice, the differentiation of muscle cells in each muscle tissue was found to be inhibited (*Bour et al., 1995*; *Lilly et al., 1995*). The *MEF2D* has also been found to be involved in skeletal myogenesis, cardiac hypertrophic growth and proliferation of vascular smooth muscle cells (*Ogawa, Sakakibara & Kamemura, 2013*; *Hu et al., 2017*; *Li et al., 2017*).

The chicken *MEF2D* gene has been cloned, but only one transcript has been reported (*Caldwell et al., 2004*). In humans and mice, multiple different transcripts of *MEF2D* gene have been found, and these transcripts can perform different functions (*Ogawa, Sakakibara & Kamemura, 2013*; *Sebastian et al., 2013*). In this study, we aim to identify the variant transcripts of chicken *MEF2D* gene from different tissue samples, measure expression levels of these transcripts in various tissues and at different periods, and to study their roles in skeletal myogenesis.

## MATERIALS AND METHODS

### Animals

The fertilized eggs of Xinghua chicken in this experiment were purchased from a livestock farm of South China Agricultural University (Guangzhou, China). They were hatched in a full-automatic incubator. During the period from the 10th embryo age (E10) to the 1st day post-hatching (P1), the breast muscle and leg muscle tissues of 20 chickens were collected each day and stored at −80 °C. Five 7-weeks-old Xinghua female chickens were purchased from a livestock farm of South China Agricultural University. A total of 15 tissues (cerebrum, cerebellum, hypothalamus, pituitary, heart, liver, spleen, lung, kidney, breast muscle, leg muscle, subcutaneous fat, abdominal fat, muscular stomach and glandular stomach) of each chicken were collected and stored at −80 °C.

### DNA samples

The DNA samples were obtained from an $F_2$ resource population crossed from Xinghua and White Recessive Rock (XH & WRR) as described previously (*Lei et al., 2005*). The population consists of 17 full-sibling families, and 434 $F_2$ individuals (221 male and 213 female chickens) with a detailed record of growth traits, carcass traits and meat quality traits. Weight (body, semi-eviscerated, eviscerated, breast muscle, leg muscle and abdominal fat pad) was measured in grams using an electronic scale. The shank length, head width, breast width, breast depth and body length were measured with vernier caliper. The shank diameter was measured in the middle of the shank with string and straightedge.

## RNA isolation, cDNA synthesis and quantitative real time PCR

Total RNA of all tissues were isolated using Trizol reagent (Invitrogen, Carlsbad, CA, USA), following the recommended manufacturer's protocol. The quality and quantity of RNA samples were assessed by gel electrophoresis and a spectrophotometer (NanoDrop 2000c; Thermo, Waltham, MA, USA). The cDNA synthesis was performed with 1 μg of RNA for each sample using a RevertAid™ First Strand cDNA Synthesis Kit (Fementas, Waltham, MA, USA) in a total reaction volume of 20 μl.

The mRNA level of MEF2D and its four variant transcripts, RBFOX2, MHC and MYOD were measured by qPCR. The qPCR was performed using SsoFast Eva Green Supermix (BIO-RAD, Hercules, CA, USA) in CFX9600 (BIO-RAD, Hercules, CA, USA). Each sample was assayed in triplicate under the following conditions: 95 °C for 2 min, followed by 40 cycles of 10 s at 95 °C, 30 s at the annealing temperature (58–62 °C), 30 s at 72 °C, a melt curve by 65–95 °C, and increments 0.5 °C for 5 s. Chicken *GAPDH* was used as the reference gene for tissue-samples of 7-weeks-old chickens and myoblasts, whereas *18S rRNA* was used as the reference gene for embryonic muscle samples. The relative mRNA level in each sample was calculated using the comparative $2^{-\Delta\Delta Ct}$ (CT is threshold cycle; $\Delta\Delta Ct = \Delta Ct_{\text{target sample}} - \Delta Ct_{\text{control sample}}$) method (*Livak & Schmittgen, 2001*).

## Gene cloning and sequences analysis

Referring to the *MEF2D* gene sequence in chicken (NM_001031600.3) reported by National Center for Biotechnology Information (NCBI), primers were designed to amplify *MEF2D* gene by PCR. Products of PCR were purified using an Agarose Gel DNA Extraction Kit (Takara, Osaka, Japan) and then cloned into the pMD-18T vector (Takara, Osaka, Japan) according to the manufacturer's protocol. Positive clones were identified by PCR and then sequenced by Invitrogen Co. Ltd (Guangzhou, China).

The sequencing results were analyzed and compared with the chicken genome (Gallus_Gallus-5.0/Galgalgal5; http://genome.ucsc.edu/cgi-bin/hgBlat) and MEF2D sequence (NM_001031600.3). DNAStar software (DNASTAR, Madison, WI, USA) was used to analyze the homology of the amino acid (AA) sequence of MEF2D between different species and the conserved regions of the sequence. The AA sequences of MEF2D from the other species were obtained from GenBank (Table S1).

## Plasmid construction, cell culture and transfection

The coding sequences of chicken *RBFOX2* and *MEF2D-V4* were amplified from cDNA of chicken leg muscle using PCR, and then cloned into the pEGFP-C1 vector (Invitrogen, Guangzhou, China) using the *EcoRI* and *BamHI* restriction sites.

Chicken primary myoblasts were isolated from the leg muscle of chickens at 10–11 embryo age as described previously (*Luo et al., 2014*). Cells were maintained in RPMI-1640 medium (Gibco, Grand Island, NY, USA) supplemented with 20% (v/v) fetal bovine serum (Gibco, Grand Island, NY, USA), and 100 μg/ml penicillin/streptomycin (Invitrogen, Guangzhou, China) at 37 °C with 5% $CO_2$, humidified atmosphere. Cells were seeded in 12-well plates with one ml per well at $10^5$ cells/ml. When the cells had grown to 70–80%

confluence, they were transfected with plasmids (one µg/ml) of *MEF2D* or *RBFOX2* or pEGFP-C1 vector control using lipofectamine 3,000 reagent (Invitrogen, Guangzhou, China) according to the manufacturer's instructions.

### Cell proliferation assay

After overexpressing *RBFOX2* and *MEF2D* genes in myoblasts for 48 h, respectively, cells were collected and fixed with 70% ethanol overnight at −20 °C. The fixed cells were collected by centrifugation at $1,000 \times g$, washed once with PBS, and stained with 0.5 ml propidium iodide (PI) dye solution (5 mg PI + 0.1 ml Triton X-100 + 3.7 mg EDTA +10 ml PBS), and then incubate for 30 min at 4 °C in the dark. After staining, cells were detected by BD FACSAriaII flow cytometer (BD, Franklin Lakes, NJ, USA). The results were analyzed by software ModFit Lt 4.1.

### Western blotting

Proteins of transfected myoblasts were extracted using RIPA lysis buffer (Beyotime, Shanghai, China) and the concentration was determined by a bicinchoninic acid protein assay kit (Beyotime, Shanghai, China). The primary antibodies MYOG (1:500 dilution; Biorbyt, Cambridge, UK) and MHC (1:1,000 dilution; DSHB, Iowa, IA, USA) were using to measure the protein levels of MYOG and MHC respectively by Western blotting as described previously (*Ouyang et al., 2018*). GAPDH (1:1,000 dilution; Bioworld, St Louis Park, MN, USA) was used as the reference gene.

### Primers

Primers were designed using primer premier 5 software (PREMIER Biosoft, Palo Alto, CA, USA) and synthesized by Bioengineering Co., Ltd. (Shanghai, China). Specific primer sequences are shown in Tables S2 and S3.

### Identification and genotyping of SNPs

Variations in the coding sequences of chicken MEF2D were identified using PCR with primers PM1–PM9 in our $F_2$ resource population (XH & WRR). The locations of primers were shown in Fig. S1. PCR was performed in 50 µl of a mixture containing 50 ng of chicken genomic DNA, 25 pmol of primers and 25 µl PCR Master Mix (Transgen, Beijing, China), and using the following protocol: 94 °C for 3 min, followed by 32 cycles of 30 s at 94 °C, 30 s at the annealing temperature (58–63 °C), 30 s at 72 °C and 72 °C for 5 min at last. Twenty DNA samples were selected randomly from the $F_2$ resource population (XH & WRR) for PCR using primers PM1–PM9. PCR products were sequenced by Bioengineering Co., Ltd. (Shanghai, China) and the results were then blasted with each other to identify variations. The special Single-nucleotide polymorphisms (SNPs) we were interested were genotyped by PCR and sequencing in all DNA samples of the $F_2$ resource population (XH & WRR).

### Statistical analysis

Single-nucleotide polymorphism frequencies were calculated using the observed numbers of alleles for each SNP. SNP genotypes were tested for Hardy–Weinberg equilibrium with

the Chi-square test. Association analysis of SNPs and fatness traits were performed using the General Linear Models Procedures of SAS 9.0 (SAS Institute Inc., Cary, NC, USA) using the following model:

$$Y_{ijkl} = \mu + S_i + G_j + H_k + F_l + e_{ijkl}$$

Where $Y$ = the traits phenotypic values; $\mu$ = the overall population mean; $S$ = the effect of gender; $G$ = the effect of genotype; $H$ = the effect of incubation batch; $F$ = the effect of family; $e$ = the random residuals.

Data on gene expression were analyzed using SPSS 21.0 (IBM, Armonk, NY, USA). The ANOVA was used to compare expression levels among different groups. All values are presented as means ± standard error of mean (SEM). The threshold for significance was set at $P < 0.05$ and for high significance at $P < 0.01$.

## Animal ethics

Animal experiments were handled in compliance and all efforts were made to minimize suffering. It was approved by the Animal Care Committee of South China Agricultural University (Guangzhou, People's Republic of China) with approval number SCAU#0014.

## RESULTS

### Sequence alignment and phylogeny analysis of MEF2D

According to the information from NCBI database, the chicken MEF2D gene cDNA sequence (NM_001031600.3) is 4,111 bp in length, the coding region is 715–2,271 nt, and it encodes 518 AAs (NP_001026771.3). Blast with the chicken genome (GRCg6a/galGal6), this gene is located on chicken chromosome 25 (2,742,900–2,782,225), the full length of the gene is 39,326 bp, and it contains 10 exons and nine introns.

The protein sequences of *MEF2D* in 10 species (*Gallus gallus*, *Meleagris gallopavo*, *Coturnix japonica*, *Homo sapiens*, *Mus musculus*, *Rattus norvegicus*, *Sus scrofa*, *Bos taurus*, *Danio rerio* and *Xenopus laevis*) were compared and analyzed by homologous clustering. The results showed that the protein sequences of *MEF2D* were highly conserved, and had conserved domain of MADS-Box (2–57 AA) and MEF2-Domain (58–86 AA) in chicken and the other nine species tested (Fig. 1A). Phylogenetic tree clustering showed that 10 species were divided into four distinct groups: birds (*Gallus gallus*, *Meleagris gallopavo* and *Coturnix japonica*), mammals (*Homo sapiens*, *Mus musculus*, *Rattus norvegicus*, *Sus scrofa* and *Bos taurus*), *Danio rerio* and *Xenopus laevis* (Fig. 1B). Homology between chicken, turkey and quail was more than 96%. Homology among mammals (human, mouse, rat, pig and cow) was also very high, while the homology between zebrafish and frogs and other species was relatively low (Fig. 1C).

### Variant transcripts of chicken MEF2D

In this experiment, cDNA samples from liver, hypothalamus and muscle tissue at different stages were used as PCR templates to clone chicken *MEF2D* gene, and positively clone PCR products were detected by agarose gel electrophoresis (Fig. 2A). Sequencing analysis of PCR products identified four novel variant transcripts (V1–V4) of *MEF2D*

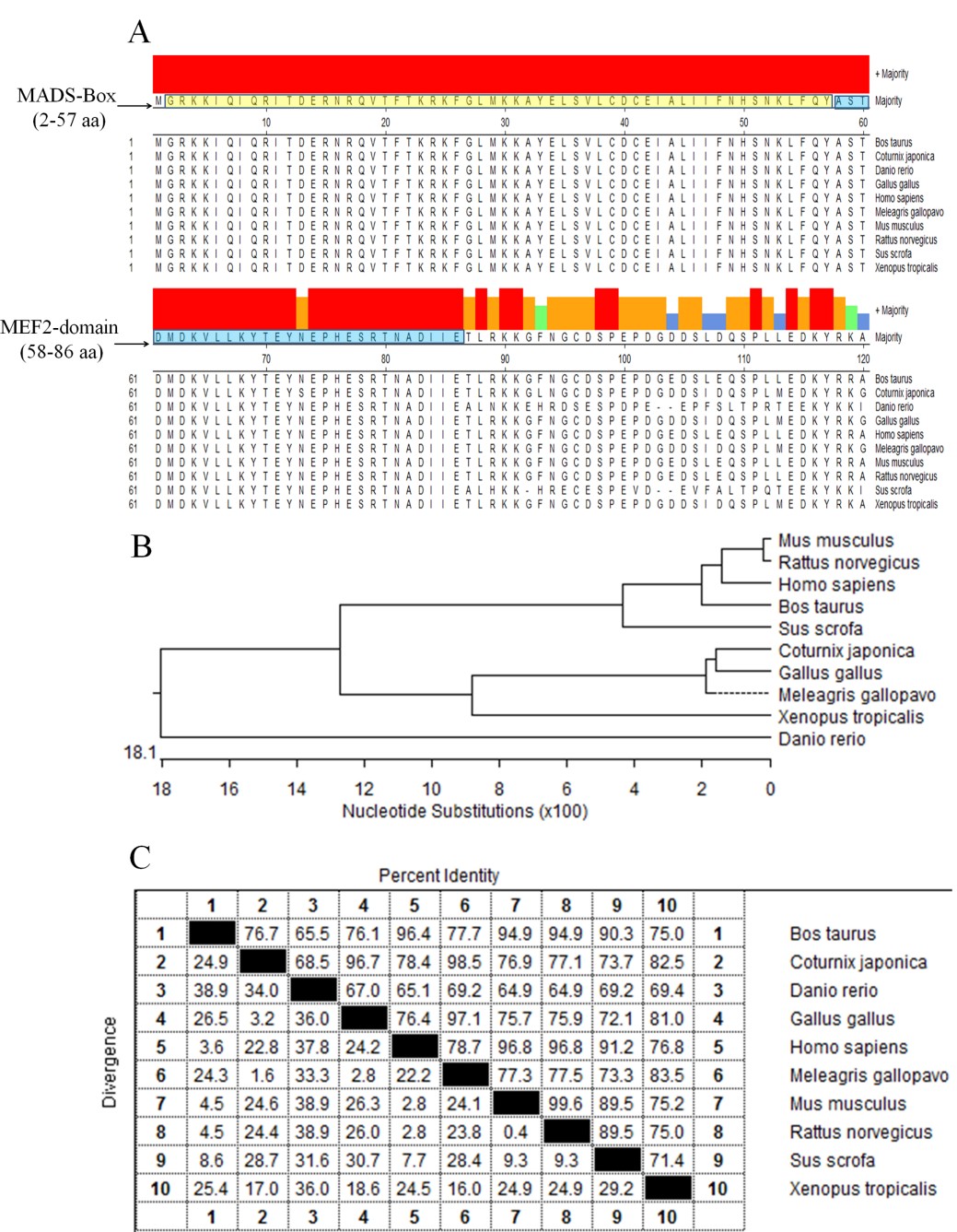

**Figure 1 Analysis of MEF2D protein sequence.** (A) The highly conserved functional region of the MEF2D protein sequence. (B) Clustering analysis of MEF2D protein sequences in ten different species. (C) Homology analysis of MEF2D protein sequences in 10 different species.

(Fig. 2B). Compared with the transcript of *MEF2D* gene on NCBI, the transcript V1 (NCBI Accession Number: KY680649) was 3,222 bp in length, had a deletion of 889 bp (1,446–2,334 nt), and was predicted to encode 251 AA. The transcript V2 (KY680650) was 3,616 bp in length, had a deletion of 498 bp (1,139–1,636 nt) and was predicted to encode 353 AA. The transcript V3 (KY680651) was 4,135 bp in length, had an insertion of

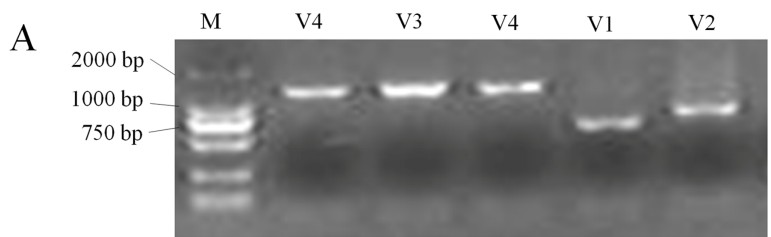

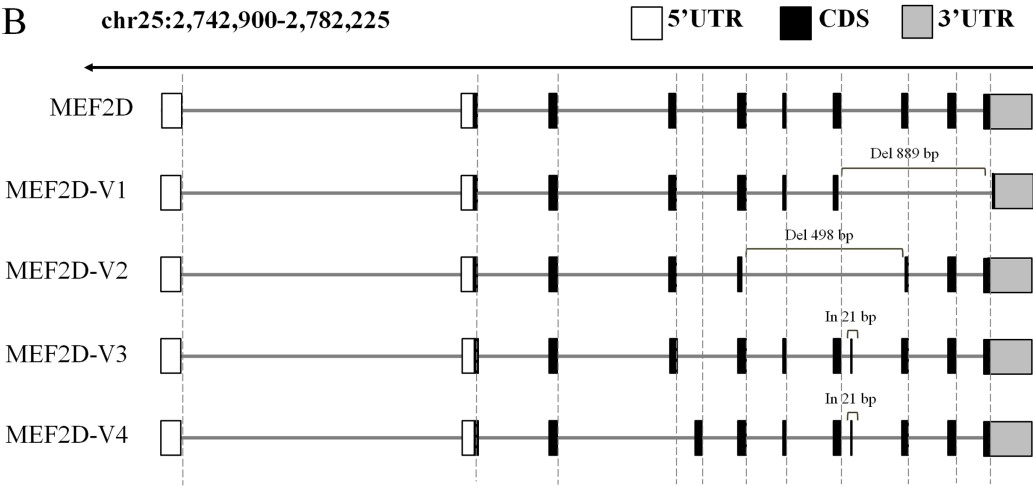

**Figure 2 Gene structures of various transcripts of chicken MEF2D.** (A) PCR amplification results of MEF2D gene cloning. (B) Gene structures of four novel transcripts. UTR, un-tranlated region; CDS, coding DNA sequence; In/Del, Insertion/deletion.

21 bp after exon 8 (1,570 nt) and an AAC insertion at 1,813 nt, and was predicted to encode 526 AA. The full length of the transcript V4 (KY680652) was 4,132 bp, and a 21 bp is inserted after exon 8 (1,570 nt), and was predicted to encode 526 AA. The complete DNA and protein sequences of these four variants are shown in File S2.

Comparative analysis of the AA sequences of the four novel *MEF2D* transcripts V1–V4 revealed that they contained conserved functional MADS-Box and MEF2-Domain. The position and sequence of exon 4 (87–132 AA) of the transcript V4 was different from that of the other transcripts. This was the same as the variant transcripts found in humans and mice, and they also mutated in the AA sequence 87–132 (Fig. S2).

## Tissue specific expression of MEF2D

The expression of *MEF2D* transcripts in the different tissues of chickens was measured. Two deletion transcripts, *MEF2D-V1* and *MEF2D-V2* were barely expressed. The main transcripts expressed were *MEF2D-1* (the same transcription as reported by NCBI), *MEF2D-V3* and *MEF2D-V4*. *MEF2D-1* and *MEF2D-V3* were expressed widely in various tissues, and the relative expression levels in adipose tissue and brain tissue were higher, and in the liver and kidney were lower (Figs. 3A and 3B). The transcript *MEF2D-V4* exhibited muscle-specific expression and was highly expressed in the heart, chest muscles and leg muscles, but its expression in other tissues was extremely low (Fig. 3C).

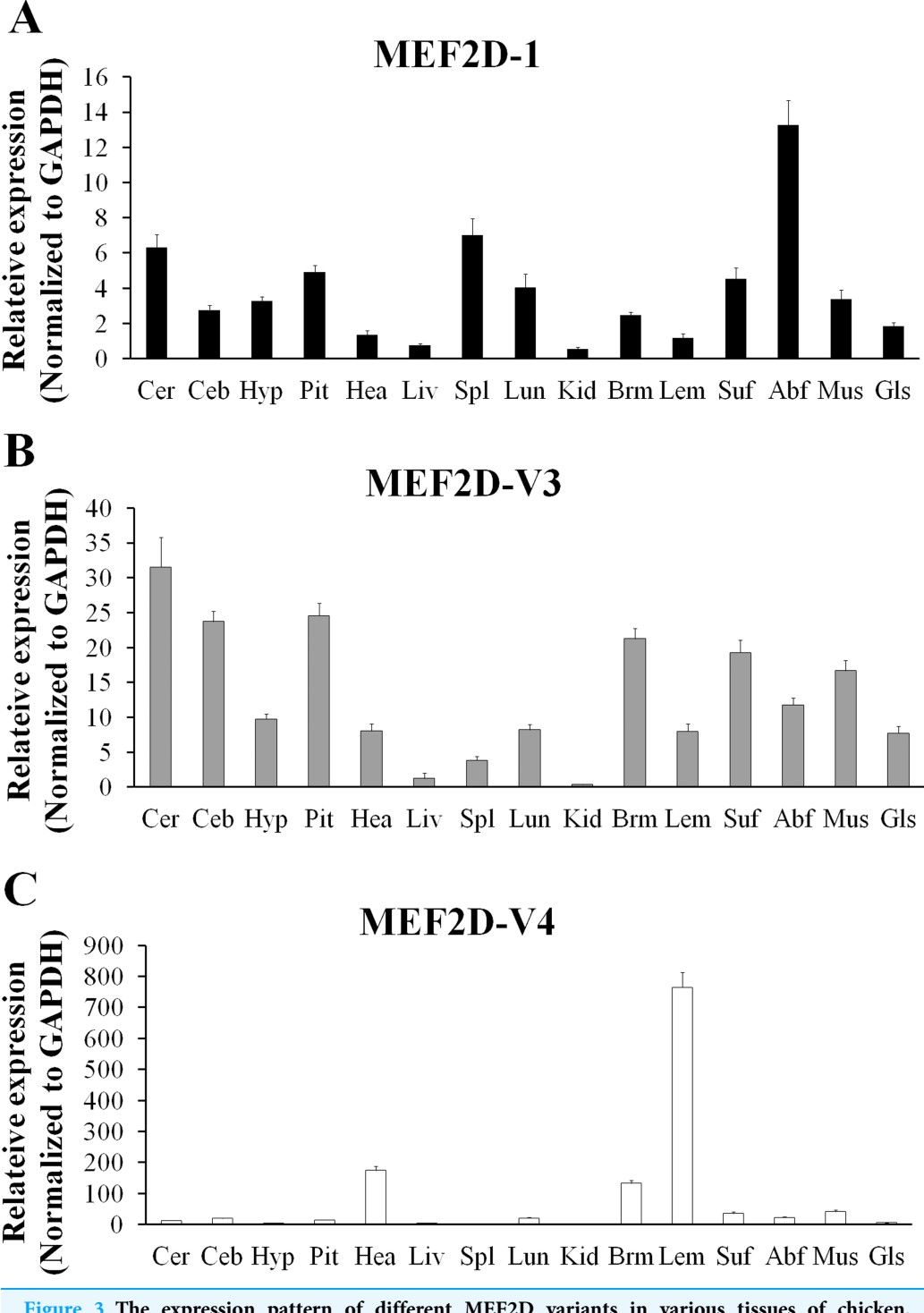

**Figure 3 The expression pattern of different MEF2D variants in various tissues of chicken.**
(A) MEF2D-1. (B) MEF2D-V3. (C) MEF2D-V4. Cer, cerebrum; Ceb, cerebellum; Hyp, hypothalamus;
Pit, pituitary; Hea, heart; Liv, liver; Spl, spleen; Lun, lung; Kid, kidney; Brm, breast muscle; Lem, leg
muscle; Abf, abdominal fat; Suf, subcutaneous fat; Mus, muscular stomach; Gls, glandular stomach.

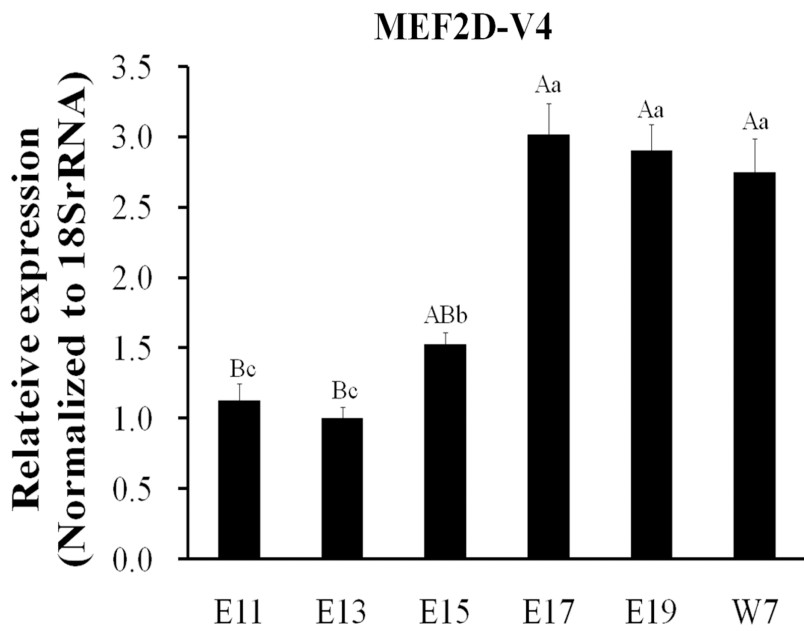

**Figure 4 Expression patterns of MEF2D-V4 in leg muscle at different stage.** Different uppercase letters on the error bar indicated extremely significant differences ($P < 0.01$), different lowercase letters indicated significant differences ($P < 0.05$), while the same letters show no significant differences ($P > 0.05$).

We measured expression level of *MEF2D-V4* in embryonic leg muscles, and found that the expression level of *MEF2D-V4* increased from E11 to E19. The expression level of *MEF2D-V4* increased significantly at E15 and E17, and it was stably expressed at E17 to E19 (Fig. 4).

## Novel transcript MEF2D-V4 promotes myoblast differentiation in chicken

The sequence of MEF2D-V4 was similar to that of the human variant transcript Mef2Dα2, expression of which was regulated by RBFOX2 and was required for muscle differentiation (*Singh et al., 2014*; *Runfola et al., 2015*). Thus, to explore the effects of MEF2D-V4 and RBFOX2 on muscle differentiation, the eukaryotic overexpression vector of *RBFOX2* and *MEF2D-V4* were constructed, and transfected into chicken myoblasts respectively. After 48 h, the expression levels of *RBFOX2* and *MEF2D-V4* were measured by qPCR: both of these two vectors could induce overexpression of the corresponding genes effectively. Furthermore, overexpression of *RBFOX2* gene could also increased the expression level of the *MEF2D-V4* significantly (Fig. 5).

After overexpressing *MEF2D-V4* and *RBFOX2* in chicken primary myoblasts, the cell cycle was detected by flow cytometry. Compared with the control group, the number of S phase cells was increased in overexpressed *MEF2D-V4* or *RBFOX2* group, but did not reach significant levels ($P > 0.05$; Fig. S3). After overexpressing *MEF2D-V4* and *RBFOX2* for 48 h in myoblast differentiation, the mRNA level of *MYOG* and *MHC* was both increased ($P < 0.05$) in cells overexpressing *MEF2D-V4* or *RBFOX2*

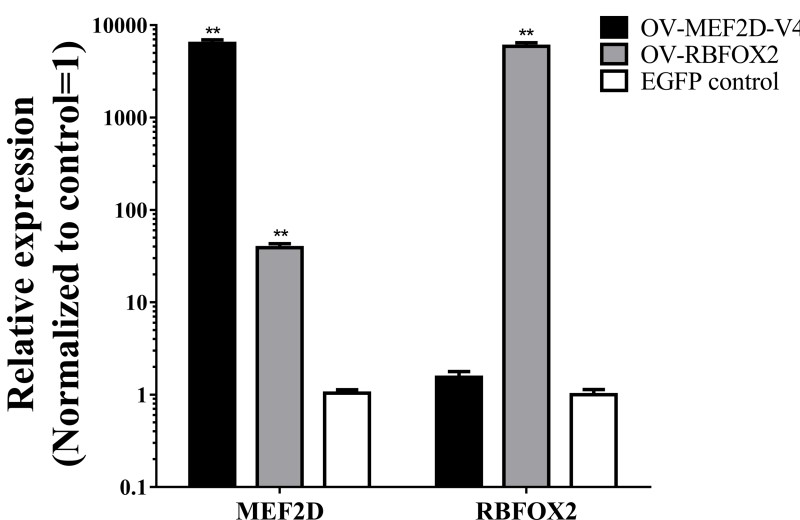

**Figure 5 Overexpression of MEF2D and RBFOX2 in chicken myoblast.** OV-MEF2D-V4 indicates overexpression vector of MEF2D-V4, OV-RBFOX2 indicates overexpression vector of RBFOX2, EGFP control indicates control vector of pEGFP-C1. **$P < 0.01$.

(Fig. 6A). The expression of *MYOG* and *MHC* was also detected by Western blot. Protein levels of MYOG and MHC were also increased, in accordance with mRNA levels (Fig. 6B).

## SNPs identification and its association analysis with production traits

In the $F_2$ resource population (XH & WRR), 31 SNPs were identified in the full length chicken MEF2D DNA through PCR sequencing (Table 1), including one insertion/deletion, 14 synonymous mutations and 16 missense mutations. Interestingly, there was a T–C mutation at exon 9, g.36186C > T, generate a TAG stop codon, resulting in a change in the coding sequence and termination of translation in both MEF2D-1 and MEF2D-V4. Therefore, we genotyped this SNP by PCR amplification and sequencing on exon 9, and carried out association analysis in the $F_2$ resource population (XH & WRR). Several growth traits were associated significantly with this SNP g.36186C > T, including first days, 7, 14, 21, 28 and 63 days of body weight, 42, 77 and 88 days of shank length, 42 and 56 days of shank diameter, and 0–4 weeks of average weight gain (Table 2). The dominant genotype of SNP g.36195C > T was the CC type, and the average early body weight of TT type individuals was lower than that of CC type individuals.

In addition, g.36094CAGIns/Del (another SNP site of exon 9) was associated with carcass traits in chickens (Table 3), including eviscerated weight (EW), leg muscle weight (LMW), abdominal fat pad weight (AFW) and small intestine length (SIL). The dominant genotype of g.36094CAGIns/Del was the Del/del type. EW and LMW of the Del/del type were lower than that of the Ins/ins type.

## DISCUSSION

*MEF2D* gene is a member of the MEF2 family and plays a key role in myogenesis (*Du et al., 2008*; *Nebbioso et al., 2009*; *Della et al., 2012*). *MEF2D* gene has several transcripts in

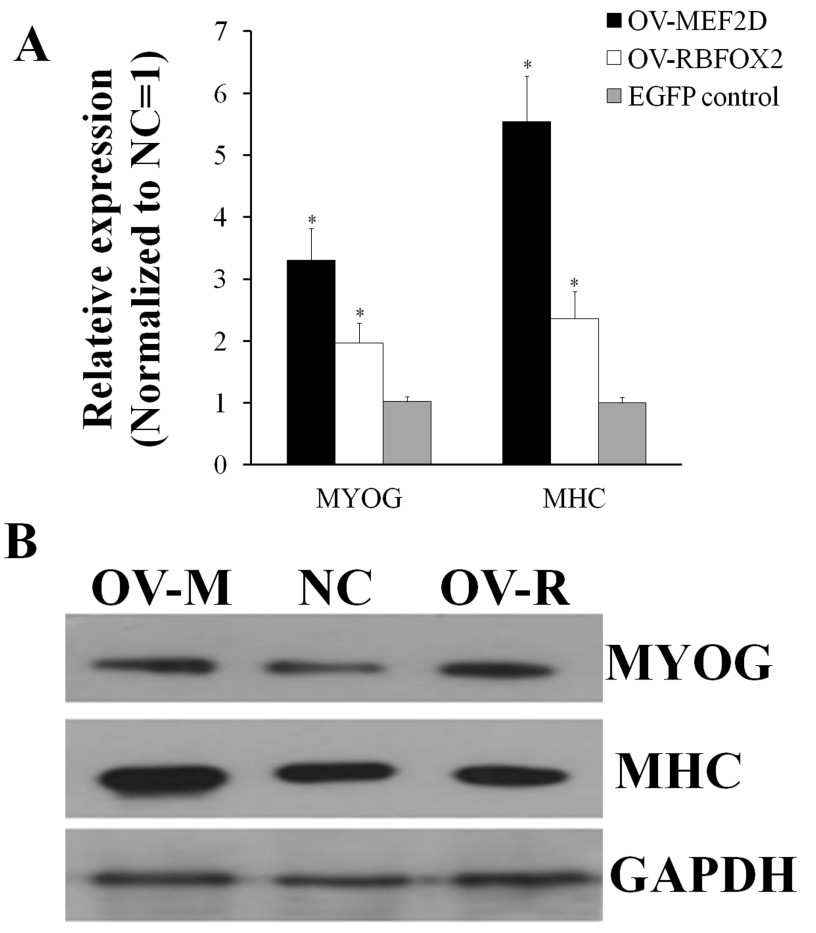

**Figure 6 Chicken MEF2D promotes primary myoblast differentiation.** (A) The expression of MYOG and MHC was determined by qPCR in primary myoblast after overexpressed MEF2D and RBFOX2. (B) The expression of MYOG and MHC was determined by Western blotting in primary myoblast after overexpressed MEF2D and RBFOX2. OV-MEF2D-V4 or OV-M indicates overexpression vector of MEF2D-V4, OV-RBFOX2 or OV-R indicates overexpression vector of RBFOX2, EGFP control or NC indicates control vector of pEGFP-C1. *$P < 0.05$.

humans and mice, and among them, there are specific transcripts that can have different functions (*Ogawa, Sakakibara & Kamemura, 2013*; *Sebastian et al., 2013*), but only one transcript sequence has been reported in chicken. Therefore, we cloned the variant transcript of the *MEF2D* gene from several different tissues of chicken, and obtained four novel transcripts. Blast analyses of their AA sequences revealed that they all contained the conserved functional regions MADS-Box and MEF2-Domain, which conformed to the structural characteristics of the MEF2 family.

The position and sequence of exon 4 of the transcript *MEF2D-V4* was different from that of the original transcript *MEF2D-1*, that is, the AA sequence 87–132 was different. The mutation region of this new transcript was similar to the variant transcript found in humans and mice, and they were both mutated in the AA sequence 87–132 region (*Edmondson et al., 1994*; *Ornatsky & McDermott, 1996*; *Nagar et al., 2017*). The human variant transcript Mef2Dα2 is expressed specifically in muscle, and it can avoid inhibitory

**Table 1  SNPs in the chicken MEF2D gene.**

| SNP name | Site in the gene | Mutation type | Note |
|---|---|---|---|
| g.26427T > G | exon5 | Missense | Thr/Pro |
| g.26446G > T | exon5 | Missense | Pro/Asn |
| g.26501G > A | exon5 | Synonymous | |
| g.26561T > G | exon5 | Synonymous | |
| g.26564C > G | exon5 | Missense | Gln/His |
| g.26590A > C | exon5 | Missense | Val/Gly |
| g.26608T > G | exon5 | Missense | Gln/Pro |
| g.26621A > C | exon5 | Missense | Ser/Arg |
| g.28390C > T | exon6 | Synonymous | |
| g.28405G > A | exon6 | Synonymous | |
| g.28423T > C | exon6 | Synonymous | |
| g.30792A > G | exon7 | Missense | Ser/Pro |
| g.30808G > A | exon7 | Missense | Pro/Leu |
| g.30852A > T | exon7 | Missense | Ser/Thr |
| g.30857A > G | exon7 | Synonymous | |
| g.30860C > G | exon7 | Synonymous | |
| g.30866T > G | exon7 | Synonymous | |
| g.30888A > G | exon7 | Missense | Ser/Pro |
| g.30892T > G | exon7 | Missense | Asn/Thr |
| g.30921T > G | exon7 | Missense | Thr/Pro |
| g.33959C > G | exon8 | Missense | Ala/Pro |
| g.36092A > G | exon9 | Synonymous | |
| g.36094CAGIns/Del | exon9 | Insert/Delete | |
| g.36137A > G | exon9 | Synonymous | |
| g.36176A > G | exon9 | Synonymous | |
| g.36179A > G | exon9 | Synonymous | |
| g.36186C > T | exon9 | Missense | Gln/stop |
| g.37162T > C | exon10 | Synonymous | |
| g.37187T > G | exon10 | Missense | Thr/Pro |
| g.37270T > G | exon10 | Synonymous | |
| g.37287T > G | exon10 | Missense | His/Pro |

phosphorylation, recruit Ash2L to activate muscle-related genes, and promote muscle cell differentiation (*Sebastian et al., 2013*). Splice variations were also found in mice, producing two transcripts, *Mef2D1a* and *Mef2D1b*. *Mef2D1a* can promote the expression of *MYOG* gene by binding to its promoter, and such binding is regulated by glycosylation (*Ogawa, Sakakibara & Kamemura, 2013*). We examined the expression patterns of these four novel transcripts, and found that *MEF2D-V4* was also expressed specifically in muscle of the heart, breast and leg. The function of muscle-specific genes is often related to muscle development and growth. During embryonic development, the expression level of *MEF2D-V4* in leg muscle was increased significantly in the late stage embryos, indicating that *MEF2D-V4* may play an important role in embryonic muscle development

**Table 2 SNP g.36186C > T associated with growth traits in chicken.**

| Traits | P-value | Least-mean-squares ± s.e.m | | |
|---|---|---|---|---|
| BW1 (g) | 0.0001 | 26.59 ± 0.64[C] (TT, 15) | 28.84 ± 5.05[B] (TC, 56) | 29.91 ± 0.16[A] (CC, 251) |
| BW7 (g) | 0.0004 | 54.90 ± 2.48[AB] (TT, 15) | 55.08 ± 1.17[B] (TC, 56) | 59.85 ± 0.54[A] (CC, 251) |
| BW14 (g) | 0.0001 | 116.62 ± 4.61[AB] (TT, 15) | 113.80 ± 2.39[B] (TC, 56) | 125.44 ± 1.13[A] (CC, 251) |
| BW21 (g) | 0.0007 | 193.15 ± 8.25[ABb] (TT, 15) | 195.66 ± 4.35[Bb] (TC, 56) | 221.01 ± 2.05[Aa] (CC, 251) |
| BW28 (g) | 0.0064 | 294.51 ± 11.95[AB] (TT, 15) | 291.62 ± 6.19[B] (TC, 56) | 312.22 ± 2.94[A] (CC, 251) |
| BW63 (g) | 0.0264 | 949.07 ± 40.02[ab] (TT, 15) | 965.01 ± 23.14[b] (TC, 56) | 1023.99 ± 11.65[a] (CC,251) |
| SL42 (mm) | 0.0159 | 59.38 ± 1.07[AB] (TT, 15) | 59.20 ± 0.58[B] (TC, 52) | 60.90 ± 0.26[A] (CC, 248) |
| SL77 (mm) | 0.0417 | 84.23 ± 1.67[b] (TT, 9) | 88.34 ± 1.06[a] (TC, 23) | 88.80 ± 0.60[a] (CC, 69) |
| SL84 (mm) | 0.0465 | 86.95 ± 1.81[ab] (TT, 9) | 90.96 ± 0.95[a] (TC, 33) | 88.63 ± 0.43[b] (CC,157) |
| SD42 (mm) | 0.0069 | 7.75 ± 0.17[AB] (TT, 15) | 7.56 ± 0.093[B] (TC, 52) | 7.89 ± 0.043[A] (CC, 248) |
| SD56 (mm) | 0.014 | 8.63 ± 0.20[AB] (TT, 15) | 8.47 ± 0.10[B] (TC, 56) | 8.79 ± 0.049[A] (CC, 248) |
| 0-4 Wks ADG (g/w) | 0.0149 | 9.57 ± 0.42[AB] (TT, 15) | 9.38 ± 0.22[B] (TC, 56) | 10.07 ± 0.10[A] (CC, 251) |

Notes:
BW, body weight; SL, shank length; SD, shank diameter; 0–4 WKs ADG (g/w), 0–4 weeks of average weight gain (g/week).
Letters and numbers in bracket refer to genotype and number of chickens with that genotype.
[a,b] $P < 0.05$.
[A,B,C] $P < 0.01$.

**Table 3 SNP g.36094CAGIns/Del associated with carcass traits in chicken.**

| Traits | P-value | Least-mean-squares ± s.e.m | | |
|---|---|---|---|---|
| EW (g) | 0.0042 | 1093.2 ± 107.3[A] (Ins/ins, 62) | 1075.5 ± 114.2[AB] (Ins/del, 80) | 1024.9 ± 102.5[B] (Del/del, 160) |
| LMW (g) | 0.013 | 119.5 ± 13.2[a] (Ins/ins, 62) | 117.4 ± 14.3[a] (Ins/del, 80) | 112.6 ± 10.8[b] (Del/del, 160) |
| AFW (g) | 0.021 | 24.33 ± 2.54[b] (Ins/ins, 54) | 28.92 ± 4.24[b] (Ins/del, 69) | 26.91 ± 4.16[ab] (Del/del, 142) |
| SIL (mm) | 0.028 | 133.6 ± 15.4[ab] (Ins/ins, 50) | 149.5 ± 11.5[a] (Ins/del, 62) | 136.2 ± 10.2[b] (Del/del, 137) |

Notes:
EW, eviscerated weight; LMW, leg muscle weight; AFW, abdominal fat pad weight; SIL, small intestine length.
Letters and numbers in bracket refer to genotype and number of chickens with that genotype.
[a,b] $P < 0.05$.
[A,B] $P < 0.01$.

and growth. Therefore, we studied further the function of *MEF2D-V4* in chicken primary myoblasts. It has been reported in mice that RBFOX2 regulates alternative splicing of the *MEF2D* gene (*Singh et al., 2014*; *Runfola et al., 2015*). We also found that chicken *RBFOX2*

promotes the expression of *MEF2D-V4*. Overexpression of *RBFOX2* and *MEF2D-V4* promoted the differentiation of chicken myoblasts.

Studies have shown that SNPs of MEF2D gene could affect the production performance of livestock and poultry animals. The MEF2D variants have been found to be highly correlated with MEF2D mRNA and protein levels in the *longissimus dorsi* muscle of cattle (*Juszczuk-Kubiak et al., 2012*). In duck, a CAG repeat polymorphism has been found in MEF2D gene. This CAG repeat can generate significantly longer transcription products and positive correlations with five muscle-related traits (*Wang et al., 2016*). We also found that a CAG insertion/deletion in MEF2D gene was associated with EW and LMW of chicken. Furthermore, g.36186C > T was found to be associated with body weight at 1, 7, 14, 21 and 28 days. This mutation generated a TAG stop codon, caused MEF2D-V4 to terminate translation early, resulting in TT type individuals not being able to produce normal MEF2D-V4 protein products. The average early body weight of TT type individuals was lower than that of CC type individuals, which indicated that MEF2D V4 may be positively correlated with chicken growth traits and promote early growth of chickens.

## CONCLUSIONS

In summary, the *MEF2D* gene can produce the muscle-specific transcript *MEF2D-V4*, which is positively regulated by *RBFOX2* and can promote the differentiation of chicken myoblasts. The chicken *MEF2D* gene could regulate the embryonic development and early growth of skeletal muscle by alternative splicing.

### Funding

This work was supported by the grants from the Science and Technology Planning Project of Guangzhou City (201504010017) and Guangdong Province (2018B020203001), and the Ten-Thousand Talents Program of China (W03020593). The funders had no role in study design, data collection and analysis, decision to publish, or preparation of the manuscript.

### Grant Disclosures

The following grant information was disclosed by the authors:
Science and Technology Planning Project of Guangzhou City: 201504010017.
Guangdong Province: 2018B020203001.
Ten-Thousand Talents Program of China: W03020593.

### Competing Interests

The authors declare that they have no competing interests.

### Author Contributions

- Hongjia Ouyang conceived and designed the experiments, performed the experiments, analyzed the data, prepared figures and/or tables, authored or reviewed drafts of the paper, and approved the final draft.

- Jiao Yu performed the experiments, analyzed the data, authored or reviewed drafts of the paper, and approved the final draft.
- Xiaolan Chen performed the experiments, prepared figures and/or tables, and approved the final draft.
- Zhijun Wang performed the experiments, prepared figures and/or tables, and approved the final draft.
- Qinghua Nie conceived and designed the experiments, authored or reviewed drafts of the paper, and approved the final draft.

## Animal Ethics

The following information was supplied relating to ethical approvals (i.e., approving body and any reference numbers):

Animal experiments were approved by the Animal Care Committee of South China Agricultural University (Guangzhou, People's Republic of China) with approval number SCAU#0014.

## DNA Deposition

The following information was supplied regarding the deposition of DNA sequences:

The sequences of MEF2D novel trancripts are available at GenBank: KY680649 to KY680652.

## Data Availability

The raw measurements are available in Files S1 and S2.

## Supplemental Information

Supplemental information for this article can be found online at http://dx.doi.org/10.7717/peerj.8351#supplemental-information.

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
