# Peer review of "A novel transcript of MEF2D promotes myoblast differentiation and its variations associated with growth traits in chicken"

_PeerJ, doi:10.7717/peerj.8351_

## Round 0.1 · original submission · Major Revisions

Dear Dr. Ouyang,

Please follow the suggestions of Reviewer #1 and #2.

Both reviewers believed that the English should be improved.

Data presentation should be improved (Reviewer #1 and #2) particularly detailed description of methodology and results of the qPCR data.
Reviewer #2 raises important issues regarding your references genes, particularly if the reported genes can be really considered reference genes. Please address that issue as well in a revised version, including proper data presentation of these aforementioned issues.

Best regards
Rodrigo

Reviewer 1 ·

Basic reporting

• Comment on languages and grammar issues.
The English language should be improved to ensure that an international audience can clearly understand the text. For example, the authors should avoid putting a coma before an “and” in an enumeration or connecting phrases. Some examples are at lines 37, 40, 59 or 107. I recommend a native English speaker review the manuscript.
• Figures
There is an error in the order of the figures. Figure S1 appear mentioned in the manuscript at line 188 and Figure S2 at line 134. The authors should change the order of the figures.

Experimental design

No comment

Validity of the findings

No comment

Additional comments

General comments for the author
Lines 84-92. Please, mention in this part the expression of which genes and transcripts are going to be quantified by RT-qPCR
Line 128. Please, add close to the software the reference or URL of the primer software.
Lines 137 – 139. In how many individuals was the MEF2D gene sequenced? In how many individuals were the SNPs genotyped?
Line 150. Please, specify the data analyzed with SPSS instead “other data”.
Line 152. As SEM is an abbreviation, please put the long name and SEM between brackets the first time that appear.
Line 174. It is template instead temple, please change the word.
Lines 177- Line 182. The sizes of the transcripts do not match with the sizes that appear in the electrophoresis gel in Figure 2B.
Lines 195-196. I do not understand what the authors mean when they say that “the expression levels in each tissue are not significantly different”. If they are talking about the expression of the same transcript in different tissues, the expression is significantly different among tissues according to the figure, for example, in figure 3A, the expression is significantly different among liver and cerebellum. Please, clarify this sentence.
Figure 4. Codification letters for significant differences according to the p-value appear in the figure caption. These codification letters are “a” and “b” or “A” and “B”. However, letter “c”, also appear in the figure, please added to the figure caption otherwise, this is confusing. In addition to this, I think there is another error, according with the error bar, the expression in E15 is different to E17, E19 and W7, however, letter A appear in this bar. Please, correct or clarify this error.

Reviewer 2 ·

Basic reporting

The author included a good introduction and discussion. However, authors should considering review the English writing. For example line 191”… two deletion transcripts MEF2D-V1 and MEF2D-V2 were hardly expressed”.
Some data are missing as the cell cycle data. Additionally the Cq values seems to be missing as well in the raw data. Since the data obtained by qPCR are an important for the author’s conclusion Cq values and melting curves are important.

Experimental design

1. The paper data relied in initial PCR results. However important information is missing to validate the data. In order to ensure the paper reportability it is important to better explain the section 2.3. For example: what was the amount of RNA used to cDNA synthesis? What method was used to calculate the efficiency of the assay to ensure that 2-ΔΔCt was an appropriated quantification method? Which RNA quality checks were used? Please verify all primer sequence, specially 18SrRNA and MYHB1. Please review the MIQE guidelines.
2. There are missing important information in other section as well. Would be important to review materials and methods section. For example: section 2.2 DNA samples the method must be better explained and the information about F2 population should be added to section 2.1 animals. In section 2.5 who many cells were plated and transfected?
3. Which tests were performed to ensure that GAPDH and 18s was indeed appropriated references genes? Specially GAPDH because it was used in a broad of different tissue to compare expression levels. Thus GAPDH expression must be stable and similar in all tissue analyzed in order allow the comparison related to figure 3.
4.Propidium iodide does not stain viable cells. Is missing information at materials and methods section 2.6 or is the data related to viability not cell proliferation? Why a representative data or results values are not shown?

Validity of the findings

The authors bring an intersting data. However it is important to better present/struture the findings.

Additional comments

The paper from Ouyang and collaborator demonstrated the presence of four transcriptor variants of MEF2D gene in chicken tissue. However, the missing of important technical information makes it difficult to analyze some data. The gene variant expression comparison between tissues and embryonic ages are crucial data. The conclusion take from these data need to be better validate.

---

## Round 0.2 · Major Revisions

Dear Dr. Ouyang,
Please follow the comments of the reviewer #2, particularly regarding your qPCR data. Please observe that most of your findings are based on comparison to reference gene expression and for this reason it is important that your data supports that your "reference gene" is a actually a reference gene.

Best regards
Rodrigo

Reviewer 1 ·

Basic reporting

No comment.

Experimental design

No comment.

Validity of the findings

No comment.

Additional comments

No comment.

Reviewer 2 ·

Basic reporting

The language was improve.

Experimental design

1. How did you check the qPCR efficiency? It's only possible to used delta delta (Livak and Schmittgen, 2001) if the assay efficiency between reference gene and investigation gene are similar (Pfaffl, 2001). Additionally, are the primers sequence of 18SrRNA and MYHB1 correct? The BLAST search of the sequence listed in the manuscript do not retry the correct products.

2. The cell density should be expressed as cell/cm2 or cell/mL and specify the volume added and the cell area (the cell area is already there as indication of 12-weel plate).

3. There is not such thing is wildly used reference gene. For example, GAPDH expression is influence by glucose, cell cycle, inflammation. Thus, a proper validation must be performed to ensure that the genes used are indeed a good reference gene. Thus, I would like to know if you looked for statistics difference in Cq values between the groups for assume gapdh ou 18s are suitably reference genes. Your most important data is based on qpcr data. We have to make sure that this data was performed correctly. For more information read Kozera and Rapacz, J Appl Genetics (2013) 54:391–406.

Validity of the findings

The manuscript has been improved. However, important information are still missing.

Additional comments

Important information is still missing.

---

## Round 0.3 · Minor Revisions

Dear Dr. Ouyang,

Although the manuscript has been largely improved, there are still some issues to be answered. Please provide detailed answers to questions #1 and #3 to the reviewer #2.

Best regards
Rodrigo

Reviewer 2 ·

Basic reporting

no comment

Experimental design

I maintained the question numbers.
1. Thank you for the explanation regarding the qPCR efficiency. The primers sequence must be checked. A search in primer blast for the sequence from the manuscript retrieves no product in the 18s primer and another product (NC_006103.5) for MYHB1 primer.
2. Ok, Thank you.
3. It is not clear if the authors tested if the chosen genes were indeed correct reference genes to be used. A simple compare between Cq could help to answer it.

Validity of the findings

no comment

Additional comments

no comment

---

## Round 0.4 · Minor Revisions

Dear Dr. Ouyang,

Before the paper can be accepted for publication please insert the full Cq values of all data as supplementary raw data.

Reviewer 2 ·

Basic reporting

The authors checked the data and corrected the missing information.

Experimental design

no comments.

Validity of the findings

I still have some doubts about the variance expression of the reference genes. However, the manuscript is valid and should be published. I strong suggest inserting the full Cq values of all data as supplementary raw data.

---

## Round 0.5 · accepted · Accept

Dear Dr. Ouyang,

Thanks for adding the raw qPCR data to the supplementary data and congratulations for the acceptance of your manuscript.